# Adipokines and Metabolic Regulators in Human and Experimental Pulmonary Arterial Hypertension

**DOI:** 10.3390/ijms22031435

**Published:** 2021-02-01

**Authors:** Aimilia Eirini Papathanasiou, Fotios Spyropoulos, Zoe Michael, Kyoung E. Joung, Despina D. Briana, Ariadne Malamitsi-Puchner, Christos S. Mantzoros, Helen Christou

**Affiliations:** 1Department of Pediatric Newborn Medicine, Brigham and Women’s Hospital, Boston, MA 02115, USA; aimilia.f.papa@gmail.com (A.E.P.); fspyropoulos@bwh.harvard.edu (F.S.); kjoung@partners.org (K.E.J.); 2Harvard Medical School, Boston, MA 02215, USA; zoe.michael@childrens.harvard.edu; 3Department of Pediatrics, National and Kapodistrian University of Athens Medical School, 10679 Athens, Greece; brianadespina@yahoo.com (D.D.B.); amalpu@med.uoa.gr (A.M.-P.); 4Department of Pediatrics, Boston Children’s Hospital, Boston, MA 02215, USA; 5Division of Endocrinology Diabetes and Metabolism, Beth Israel Deaconess Medical Center, Boston, MA 02215, USA; 6Section of Endocrinology, VA Boston Healthcare System, Harvard Medical School, Boston, MA 02215, USA

**Keywords:** pulmonary hypertension, adipose tissue, meta-inflammation

## Abstract

Pulmonary hypertension (PH) is associated with meta-inflammation related to obesity but the role of adipose tissue in PH pathogenesis is unknown. We hypothesized that adipose tissue-derived metabolic regulators are altered in human and experimental PH. We measured circulating levels of fatty acid binding protein 4 (FABP-4), fibroblast growth factor -21 (FGF-21), adiponectin, and the mRNA levels of FABP-4, FGF-21, and peroxisome proliferator-activated receptor γ (PPARγ) in lung tissue of patients with idiopathic PH and healthy controls. We also evaluated lung and adipose tissue expression of these mediators in the three most commonly used experimental rodent models of pulmonary hypertension. Circulating levels of FABP-4, FGF-21, and adiponectin were significantly elevated in PH patients compared to controls and the mRNA levels of these regulators and PPARγ were also significantly increased in human PH lungs and in the lungs of rats with experimental PH compared to controls. These findings were coupled with increased levels of adipose tissue mRNA of genes related to glucose uptake, glycolysis, tricarboxylic acid cycle, and fatty acid oxidation in experimental PH. Our results support that metabolic alterations in human PH are recapitulated in rodent models of the disease and suggest that adipose tissue may contribute to PH pathogenesis.

## 1. Introduction

Pulmonary Arterial Hypertension (PAH), a progressive and often fatal disease that primarily affects the pulmonary circulation and right ventricle of the heart, is increasingly recognized as a systemic disorder in which inflammatory and metabolic derangements may play a pathogenetic role [1,2,3]. Obesity and associated metabolic disorders such as insulin resistance, type II Diabetes, and cardiovascular diseases are associated with both idiopathic and secondary forms of pulmonary hypertension (PH) [4,5,6,7,8] and data suggest that they modulate disease onset and severity [9]. Meta-inflammation, the low grade chronic inflammatory state triggered by metabolic imbalances is often seen in metabolic syndrome, but its role in PAH pathogenesis is not completely understood. Meta-inflammation is thought to be mediated by immune cells in extrapulmonary tissues, such as the colon, liver, skeletal muscle, and adipose tissue [10]. A better understanding of the role of adipokines and meta-inflammation in PAH pathogenesis will provide the basis for therapeutic approaches that may confer benefit in these patients.

Adipose tissue, a metabolically active endocrine organ, which is implicated in cardiometabolic homeostasis, may be involved in PAH pathogenesis through production and secretion of bioactive mediators known as adipokines that act locally and systemically. These mediators may regulate several physiologic functions including, but not limited to, glucose and lipid metabolism, insulin sensitivity, and inflammation [11]. Further to previous reports referring to the role of specific adipokines in PAH [12,13] we assumed the existence of adipose tissue metabolic implication in PAH.

In order to investigate the above hypothesis, we studied a panel of primarily adipose tissue-derived or therein implicated metabolic regulators including Fatty acid-binding protein-4 (FABP-4), Fibroblast growth factor-21 (FGF-21), adiponectin, and the related transcription factor peroxisome proliferator-activated receptor γ (PPARγ) in human and experimental pulmonary hypertension in rats and evaluated the metabolic state of adipose tissue by assessing mRNA levels of genes involved in glycolysis, the tricarboxylic acid cycle (TCA), and fatty acid oxidation (FAO) in experimental pulmonary hypertension [1].

## 2. Results

### 2.1. Circulating Concentrations of FABP-4, FGF-21, and Adiponectin Are Significantly Elevated in Patients with Idiopathic Pulmonary Arterial Hypertension (IPAH) Compared to Healthy Controls

IPAH patient and control group demographic, somatometric, and clinical characteristics are summarized in Appendix A. Circulating levels of the predominantly adipose tissue-derived cytokines FABP-4 (*p* = 0.0107), FGF-21 (*p* = 0.0013), and adiponectin (*p* = 0.0021) were significantly increased in the serum of patients with IPAH, compared to the normal control group (Figure 1). Statistically significant positive correlations were documented between serum FGF-21 and FABP-4 levels (*p* < 0.0001, *r* = 0.612), adiponectin and FABP-4 levels (*p* = 0.014, *r* = 0.402) and between adiponectin and FGF-21 levels (*p* = 0.029, *r* = 0.37).

### 2.2. Lung Levels of FGF-21, FABP-4, and PPARγ mRNA Are Significantly Elevated in Patients with IPAH Compared to Healthy Controls

Given that the expression of FABP-4, FGF-21, and the related transcription factor PPARγ were previously reported in non-adipose tissues, we evaluated their mRNA expression in the lungs of IPAH patients and normal controls. As shown in Figure 2, we found that PAH was associated with significantly increased lung mRNA levels of FABP-4 (*p* = 0.0004), FGF-21(*p* = 0.001), and the transcription factor PPARγ (*p* = 0.0032), when compared to control subjects (failed donors). We further evaluated the expression of FABP-4 in pulmonary arteries and found no differences between IPAH subjects and controls (data not shown).

### 2.3. Pulmonary Hypertension Was Induced in All Three PH-Experimental Models

We performed hemodynamic measurements in anesthetized rats. The Right Ventricular Systolic Pressure (RVSP), which provides an estimate of blood pressure in the pulmonary circulation, was significantly elevated in all three PH-experimental models compared to the normoxic controls. (Chronic Hx: 37.03 ± 1.5 mm Hg, *p* = 0.003; Sugen/Hypoxia (SuHx): 38.91 ± 2.3 mm Hg, *p* < 0.0001; Monocrotaline (MCT): 52.18 ± 8.8 mm Hg, *p* = 0.03.) We did not find any difference in Left Ventricular Systolic Pressure (LVSP), an indicator of blood pressure in the systemic circulation, in rats with experimental PH compared to normoxic animals (Table 1). In addition, chronic hypoxia, MCT and SuHx rats exhibited right ventricular (RV) hypertrophy, as assessed by Fulton’s Index (FI) and right ventricular (RV) to total body weight (BW) (RV/BW) ratio, which were significantly increased compared to the control animals (Table 1).

### 2.4. Lung Levels of FGF-21, FABP-4, and PPARγ mRNA Are Significantly Elevated in Experimental Models of PH

To assess whether changes in lung adipokine mRNA levels seen in human IPAH are recapitulated in experimental PH, we evaluated mRNA levels of FABP-4, FGF-21, and PPARγ in rat lungs and found them significantly upregulated in all three models of experimental PH (Figure 3). We did not detect adiponectin mRNA in lungs but found significantly increased mRNA levels of adiponectin receptors (R) (1 and 2) in the SuHx model. Significantly increased circulating FABP-4 levels were also found in the SuHx model (data not shown).

### 2.5. Altered Expression of PPARγ and FABP-4 in Adipose Tissue in Experimental PH

To assess the potential contribution of adipose tissue in experimental PH pathogenesis, we evaluated the expression levels of PPARγ, the master regulator of adipogenesis and insulin responsiveness in adipose tissue and found increased mRNA levels in all three experimental models of PH (Figure 4). mRNA expression of PPARα was significantly increased in the MCT model. In addition, the downstream target of PPARγ and FABP-4 was also upregulated in adipose tissue in all models of PH. Levels of adiponectin, Adiponectin R1, and Adiponectin R2 were significantly increased in the MCT and SuHx model (Figure 5). In the hypoxic model, there was a trend towards increased mRNA expression of Adiponectin R1 and R2 (*p* = 0.054), but this did not reach statistical significance. Levels of FGF-21 mRNA levels were not significantly changed in any of the three experimental models (data not shown).

### 2.6. Metabolic Pathway Gene Expression Is Altered in Rat Adipose Tissue in Experimental PH

To evaluate the metabolic state of adipose tissue during experimental PH we assessed mRNA levels of genes involved in glycolysis, the TCA and FAO. Glucose uptake and glycolysis: as shown in Figure 6, levels of Glucose transporter 4 (Glut-4) mRNA, the glucose transporter that facilitates insulin-stimulated glucose uptake, were significantly increased in the SuHx and MCT models and a non-significant trend was noted in the hypoxic model. Interestingly, insulin-independent Glut-1 mRNA expression was only significantly increased in the hypoxic model, while no significant difference was determined in the SuHx- or MCT-induced PH. However, mRNA levels of Phosphofructokinase-1(PFK-1), one of the key regulatory and rate limiting steps in glycolysis were significantly increased in adipose tissue in all three models of PH.

TCA: Pyruvate dehydrogenase beta subunit catalyzes the conversion of pyruvate to acetyl-CoA and CO_2_, thus linking the glycolytic pathway to the TCA cycle and its activity is inhibited by Pyruvate Dehydrogenase Kinase (PDK). As shown in Figure 7, pyruvate dehydrogenase beta mRNA levels were elevated in the MCT and SuHx models with a non-significant trend in the hypoxic model. We observed similar changes in the mRNA expression of citrate synthase, which catalyzes the first step in the TCA cycle and PDK-4.

FAO: We evaluated mRNA levels of CD36, which facilitates net fatty acid uptake in adipose tissue and found them significantly upregulated in all the three established PH-experimental models, when compared to normal controls (Figure 8). In addition, expression of acylCoA dehydrogenases for short, medium, and very long-chain fatty acid substrates, which catalyze the initial step of β-oxidation in mitochondria, were all significantly increased in the MCT model with similar trends in the other two models. In addition, we found significantly increased Acetyl-CoA carboxylase-2 mRNA levels in the Su/Hx and MCT model and a non-significant trend in the hypoxic model.

## 3. Discussion

In this study, we found altered adipose tissue secretory pattern characterized by increased circulating levels of FABP-4, adiponectin, and FGF-21 in patients with IPAH compared to healthy controls, as well as a significant positive correlation of serum FABP-4 with FGF-21 levels in IPAH, accompanied by an increase in FABP-4 and FGF-21 mRNA expression in lungs from patients with PAH. In addition, we demonstrated, for the first time, that adipose tissue from rats with experimental PH (SuHx and MCT-induced and to a lower extent hypoxia-induced) exhibit altered adipose tissue bioenergetics compared to the control group. More specifically, this metabolic alteration is characterized by increased mRNA expression of enzymes involved in glycolysis, TCA cycle, and FAO, accompanied by an increase in adipose tissue mRNA expression of adiponectin and its receptors, PPARγ and FABP-4.

The pathophysiology of PAH is complex and multifactorial. The metabolic theory in PAH proposes that metabolic dysregulation in the pulmonary vasculature and extra-pulmonary tissues plays a crucial role in PAH pathogenesis [1,14,15]. Although prior descriptive and interventional studies have evaluated the role of adipokines in PAH, the role of adipose tissue in both elaborating adipokines and participating in altered metabolic state during PAH has not been previously studied. Our findings support that adipose tissue may be involved in experimental pulmonary hypertension.

FABP-4, primarily secreted by adipose tissue and to a lesser extent by macrophages, has emerged as a promising biomarker at the intersection of metabolism and inflammation. Besides, it is well described role as a lipid chaperone and adiposity biomarker, FABP-4 is associated with features of the metabolic syndrome, insulin resistance, atherosclerosis, and low-grade inflammation and has been proposed as a biomarker for cardiovascular disease and heart failure [16]. We report, for the first time, increased serum FABP-4 levels in IPAH patients compared to controls and significant positive correlations between circulating FABP-4, FGF-21, and adiponectin levels. The positive correlation between peak FGF-21 levels and peak FABP-4 levels was previously described by Sunaga et al. in patients with acute myocardial infarction [17]. FABP-4 expression is increased under the influence of sympathetic activation, which has been described in PAH and might contribute to the increased levels observed in our study [18,19]. In addition, hypoxia stimulates adipocyte FABP-4 secretion in vitro [20] and intermittent hypoxia increases subcutaneous adipose tissue FABP-4 expression in rats [21]. In humans, elevated circulating FABP-4 levels are associated with obstructive sleep apnea and significantly correlate with hypoxemia [22]. Thus, we speculate that the elevated circulating levels of FABP-4 in IPAH patients in our study may be related to the hypoxic response of adipose tissue in PAH [23].

Shields et al. reported increased FABP-4 protein levels in cardiac adipose tissue in the SuHx rat model of PH and suggested that there is adipose tissue dysfunction, which might affect vascular remodeling in an autocrine and paracrine manner for coronary and lung vessels, respectively [24]. Additionally, increased non-adipose tissue-derived FABP-4 secretion may also contribute to increased circulating levels observed in our study and our finding of increased FABP-4 mRNA in lung tissue from patients with IPAH supports this notion. Although we have not identified the precise cellular lung sources of increased FABP-4 mRNA in human IPAH, we did not find elevated levels in isolated IPAH pulmonary arteries, therefore we speculate that the source of FABP-4 is not the pulmonary vascular compartment.

Adiponectin is abundantly secreted from adipose tissue and has been extensively studied in PH [12]. Loss and gain of function studies in mice support that Adiponectin has a protective role in PH pathogenesis that is mediated by multiple mechanisms, including inhibition of inflammatory and proliferative pathways [25,26] and this is similar to the protective role that has been described for the adipokine omentin [27]. Our finding of increased serum adiponectin levels in patients with IPAH compared with healthy controls is consistent with the results of two other studies. Interestingly, in the study by Santos et al. [28], higher adiponectin levels were reported in patients with worse functional status (New York Heart Association- NYHA class III and IV), when compared with patients with less severe disease (NYHA class I and II). Our study was not powered to detect an association with disease severity. Although this finding is counterintuitive, it is in line with the recently described ‘adiponectin paradox’ characterized by a paradoxical association of high serum adiponectin levels with increased cardiovascular mortality rate [29]. This phenomenon may be related to a compensatory response of adipose tissue to adiponectin resistance state due to the down-regulation and/or phosphorylation of adiponectin receptor, which have been documented in cardiovascular disease [30]. Alternatively, the increased endogenous levels of adiponectin may be insufficient to exert the protective effects observed at supraphysiologic levels. In our study, we should take into consideration potentially confounding effects of PAH and other medications on adiponectin levels.

The FGF-21-adiponectin axis implicated in the regulation of glucose and lipid metabolism, is also related to cardiovascular homeostasis [31]. Prior studies support an obligatory role of adipose tissue in mediating certain biological actions of FGF-21, and adiponectin as the key downstream mediator of this hormone [29]. FGF-21-induced expression and secretion of adiponectin has been described in vivo and in vitro [32] and these two mediators share multiple cardiometabolic beneficial effects. It is thus possible that the increased adiponectin levels observed in patients with IPAH, compared to healthy controls in our study may be due to the increased serum FGF-21 levels observed in the same patients.

Studies in human subjects have identified increased FGF-21 levels in cardiometabolic pathologies such as non-alcoholic fatty liver disease, coronary artery disease, diastolic heart failure, diabetes, and insulin resistance, mediating the adaptive response to various stresses [33]. Increased FGF-21 levels were also proposed as a biomarker of mitochondrial dysfunction in skeletal muscles [34]. To our knowledge, no prior studies have evaluated circulating FGF-21 levels in human IPAH. We found increased FGF-21 levels in IPAH patients compared to controls. Similar to our adiponectin findings this may be due to FGF-21 resistance that has been proposed in various cardiometabolic diseases and/or as a protective compensatory response. Although, under normal physiologic conditions the liver is the primary source of circulating FGF-21 [35], in response to a variety of stressors, ectopic FGF-21 expression, and secretion from muscle, brown and white adipose tissue also occurs [36,37,38] and could account for the increased FGF-21 levels in our study. We did indeed demonstrate increased FGF-21 mRNA expression in lung tissue from patients with IPAH, but whether this contributes to the elevated circulating FGF-21 levels, rather than simply acting in an autocrine manner, remains to be elucidated. Our finding in human IPAH is in contrast with the results of two preclinical studies which demonstrated that hypoxia-induced PH in mice or rats was associated with decreased FGF-21 mRNA expression in lungs and pulmonary arterioles [39] or decreased circulating FGF-21 serum levels, respectively [40]. Interventional studies showed that exogenous FGF-21 attenuated hypoxia-induced PH and inflammatory cytokine secretion through enhancing PPARγ expression in rats [40], attenuated hypoxic pulmonary arterial remodeling and collagen deposition in mice [39], and ameliorated hypoxia-induced dysfunction in human pulmonary arterial endothelial cells through alleviating endoplasmic reticulum stress [41]. We speculate that this may be related to the fact that hypoxia-induced pulmonary hypertension in rodents may not fully recapitulate the features of human disease and/or FGF-21 resistance status mentioned above.

To delineate the metabolic impact of this dysregulated pattern of adipocytokine secretion in human and experimental PH, we evaluated adipose tissue metabolic profile in the three most commonly used experimental models of PH, MCT, hypoxia, and SuHx. We report for the first time metabolic alteration in adipose tissue in experimental PH, characterized by increased mRNA levels of genes involved in glucose uptake and glycolysis. We found increased Glut-1 adipose tissue mRNA expression in the hypoxic model of PH and this agrees with microarray and proteomic studies by Trayhurn et al. that showed a major induction in Glut-1-mediated glucose uptake in adipose tissue, as well as in key enzymes involved in the glycolytic pathway, including PFK-1 under hypoxia [42]. Our finding of increased Glut-4 adipose tissue mRNA expression, which facilitates insulin-stimulated glucose uptake in the SuHx and MCT model agrees with a study that demonstrated similar findings of increased mRNA and protein level in the RV of rats with MCT-induced PH [43]. In addition, we observed significantly increased adipose tissue mRNA levels of PFK-1, which is a key enzyme in glycolysis. Increased levels of PFK-1 protein were also reported by Calvier et al. in pulmonary arteries from idiopathic IPAH patients and were shown to induce human pulmonary artery smooth muscle cell (PASMC) proliferation [44].

When we evaluated the TCA cycle we found increased adipose tissue mRNA expression of PDH and citrate synthase, which catalyzes the first committed step in the TCA cycle. While most data in the literature suggest that the inhibition of PDH, which prevents pyruvate from entering into TCA cycle and therefore diminishes oxidative phosphorylation, mediates the metabolic dysfunction in PAH [1], metabolomic studies in lungs from patients with severe PAH have indicated upregulated TCA cycle activity, disrupted glycolysis, and increased FAO [45]. The latter is in contrast to results of a similar metabolomic study in the serum of PAH patients with less severe disease [46]. Our results suggest increased TCA cycle function in adipose tissue in the MCT and SuHx-induced PH models.

The role of FAO in PAH pathogenesis has mostly been studied in the right ventricle and relates to the development of heart failure. Reduced FAO has been documented in RV of bone morphogenetic protein receptor type II (BMPR2) mutant mice [47] and in the RV of SuHx rats. Interestingly, the PPARγ agonist pioglitazone reversed PH and right heart failure in the SuHx model of PH via increasing FAO [48]. In a model of compensated RV failure induced by pulmonary artery banding there was increased FAO and decreased GO in the RV [49]. In our models of experimental PH, we found increased adipose tissue mRNA levels of genes encoding enzymes involved in FAO such as CD36, which facilitates plasma membrane fatty acid transport and acylCoA dehydrogenases, which catalyze the initial step of β-oxidation in mitochondria. In addition, adipose tissue mRNA expression of the FAO-driving gene FABP-4 that regulates free fatty acids transport and lipid storage was significantly increased in all three experimental models of PAH. Our findings support a concomitant enhancement of both GO and FAO in contrast to the Randle cycle hypothesis where there is reciprocal relationship between glucose and fatty acid oxidation. Indeed, under hemodynamic/metabolic stress, such as under adrenergic activation and oxygen deprivation, the inhibition of carbohydrate oxidation by fatty acids is abolished [50] and this is likely mediated by 5′ adenosine monophosphate-activated protein kinase (AMPK) activation. Once activated, AMPK stimulates catabolic (FAO, Glycolysis-Glucose uptake) and switches off anabolic pathways (lipid and protein synthesis), which may explain our findings [50]. We speculate that elevated circulating adiponectin and FGF-21 levels in patients with IPAH may activate AMPK signaling in adipose tissue [51,52]. Given that we observed increased mRNA for adiponectin and its receptors, we suspect increased adiponectin autocrine action in adipose tissue in PH. In addition, FGF-21 stimulates PPARγ transcriptional activity [53], which in turn regulates adiponectin gene transcription [54]. Indeed, PPARγ, a ligand-activated transcription factor, implicated in adipogenesis and glucose metabolism [55] regulates adiponectin and other genes associated with PAH, and has been suggested as a possible and therein useful therapeutic agent [56]. PPARγ is expressed in endothelial cells of small vessels and studies support that low levels of PPARγ contribute to pulmonary vascular remodeling and plexiform lesions in PAH patients [57]. Several reports refer to decreased PPARγ expression in the PAH phenotype which is reversed after PPARγ activation [48]. The PPARγ agonists rosiglitazone and pioglitazone exert beneficial effects in several aspects of PH including: suppression of pro-inflammatory cytokines such as TGF b [58] and metalloprotease [59], suppression of angiotensin-II induced hypertension, and reduction of endothelin-1 levels [48,60,61,62]. In contrast, in our experimental PH study, the observed increased PPARγ mRNA expression might be attributed to the compensatory mechanism to overcome PAH or, a secondary phenomenon due to increased adiponectin expression.

There are several limitations to our descriptive study that raises interesting possibilities but does not provide sufficient mechanistic insights. Our observed changes in mRNA levels may not always correlate with changes in protein abundance. Our results from the human studies need to be interpreted with consideration of the potential confounding effects of patient sex, body mass index (BMI), diabetes status and additional morbidities, and medication use of the subjects. In addition, every preclinical model of PH has inherent limitations [63,64,65] and our studies at the endpoint of each model cannot provide mechanistic insights into the role of these adipokines in disease development and progression. Furthermore, although rodent models have proven useful for exploring common pathophysiologic mechanisms and performing proof-of-concept mechanistic and intervention studies, it is important to recognize that human disease is heterogeneous and thus results from preclinical studies are not generalizable to all groups of PAH. We further acknowledge that our study provides a set of observations that need to be further explored with mechanistic studies in the different experimental models of pulmonary hypertension to define the contribution of adipose tissue-derived cytokines and bioenergetics to disease pathogenesis, progression, and response to treatments. We have included several speculative statements in an effort to interpret our findings in the context of the existing literature.

In summary, we report a pattern of dysregulated adipokine circulating levels in humans with idiopathic pulmonary arterial hypertension with concomitant increased expression in the lungs. Although there are multiple potential sources of these adipokines, our findings in adipose tissue in three experimental models, support that adipose tissue may be contributing to PAH pathogenesis via adipokine release and altered bioenergetics. Consistent with the notion that PAH is a systemic disease, we report for the first time adipose tissue metabolic alterations in experimental models. Additional studies to identify the cellular sources of these adipokines, both within the lung and within adipose tissue, are needed in order to gain mechanistic insights into their potential role in disease pathogenesis.

## 4. Materials and Methods

### 4.1. Human IPAH and Control Samples

Human serum, pulmonary arteries, and lung samples were obtained through the Pulmonary Hypertension Breakthrough Initiative (PHBI, https://www.ipahresearch.org/services.html), a multicenter network of lung transplant centers created to collect tissues and biological fluids from patients with idiopathic pulmonary arterial hypertension (IPAH) listed for lung transplantation and controls (failed organ donors). Peripheral lung tissues were snap-frozen and preserved in RNA later. Blood serum was collected prior to transplantation and stored at −80 °C until assay. The control serum samples were accrued through the Partners Biobank service. All samples were de-identified, coded and matched for age and gender as closely as possible. Patient information is provided in Appendix A. Our study was approved by the Institutional Review Board (Partners Human Research Committee).

### 4.2. Animals and Experimental Models of Pulmonary Hypertension

All experimental procedures were conducted in accordance with the guidelines of the American Physiologic Society and the National Institutes of Health and were approved by the Harvard Institutional Animal Care and Use Committee and the Harvard Medical Area Standard Committee on Animals (Brigham and Women’s Institutional Animal Care and Use Committee). Adult (12-week-old) male Sprague-Dawley rats, (Charles River Laboratories, Wilmington, MA, USA) were housed in the animal facility at 22 °C ambient temperature with 12-h light/dark cycle and left to acclimatize for 2–3 days before any experimental procedure. All animals received ad libitum normal Purina Rodent Chow (Purina, St. Louis, MO, USA) and water.

Three commonly used and well described pulmonary hypertension (PH)-experimental models were utilized in this study (Schema-Appendix A): (i) Monocrotaline Model (MCT): A single subcutaneous injection of 60 mg/kg MCT (Sigma, St. Louis, MO, USA) was administered and rats were assessed for PH development 4 weeks after the injection. Control animals received the same volume of vehicle (normal saline), (ii) Sugen/Hypoxia rat model: Rats were administered a single subcutaneous injection of 20 mg/kg Sugen 5416 (Sigma, St. Louis, MO, USA) and were exposed to hypoxia at 9% O_2_ in a chamber under the control of an Oxycycler controller (BioSpherix, Redfield, NY, USA) as previously described [66,67,68]. After three weeks of exposure to hypoxia, animals were returned to normoxia (room air at 21% O_2_) for three days and then sacrificed. Control animals received the same volume of vehicle (dimethyl sulfoxide (DMSO) and were kept in normoxia. (iii) Chronic Hypoxia model: Rats were exposed to hypoxia (as described above) for 2 weeks. Control animals were kept in normoxia.

### 4.3. Hemodynamic Measurements.

RVSP and LVSP were measured as we previously published [69]. Briefly, post-anesthesia (2–3% isoflurane) inhalation, a tracheostomy was performed, and rats remained mechanically ventilated on a rodent ventilator (Harvard Apparatus, Holliston, MA, USA, tidal volume 1 mL/100 g body weight, 60 breaths/min). The diaphragm was exposed through a small transverse incision in the abdominal wall. A 23-gauge heparinized butterfly needle with tubing connected to a pressure transducer was inserted initially into the right and then into the left ventricle. Pressure measurements were obtained and recorded, using PowerLab monitoring hardware and software (ADInstruments, Colorado Springs, CO, USA). Mean RVSP and LVSP over the first 10 stable heartbeats were recorded. Over anesthetized rats presenting with heart rates less than 300 beats per minute were excluded.

### 4.4. Right Ventricular Weight and FI

Whole heart was harvested, and ventricles were isolated and weighed. Right ventricular hypertrophy was evaluated as FI, [FI: ratio of right ventricular weight (RV) to the left ventricular (LV) + septum weight] or as the ratio of RV/BW.

### 4.5. Rat Tissue Isolation

Post-hemodynamic measurements and under deep anesthesia, rats were sacrificed by exsanguination and organs were harvested, immediately. Lungs were isolated as previously described [69] and perirenal fat pads were carefully dissected from visible blood vessels. Samples were snap-frozen and stored at −80 °C until processing.

### 4.6. Gene Expression Analysis

Visceral (perirenal) adipose tissue and lung samples were harvested for RNA isolation. Total RNA was extracted using the TRIzol (Invitrogen, Carlsbad, CA, USA) method as per the manufacturer’s guidelines. RNA quantity and quality were determined by NanoDrop 2000c spectrophotometer (NanoDrop Technologies, Wilmington, DE, USA). For cDNA synthesis, samples were first treated with DNase I (Sigma, St. Louis, MO, USA) for genomic DNA digestion and then the mRNA was reverse transcribed using the Superscript III (Invitrogen Life Technologies, Carlsbad, CA, USA) reagent. Real time quantification polymerase chain reaction (RT-qPCR) was performed with a StepOnePlus thermal cycler (Applied Biosystems, Foster City, CA, USA) using either the iTaq Universal SYBR Green Supermix (Bio-Rad, Hercules, CA, USA) or the PowerUp™ SYBR^®^ Green Master Mix kit. Primers were designed using nBLAST (National Center for Biotechnology Information, Bethesda, MD, USA) and PrimerQuest Tool (Integrated DNA Technologies, Coralville, IA, USA). The comparative Ct method (ΔΔCT) was used to calculate the relative gene expression level [70]. 18S was used as the reference gene. Primer sequences used in this study are presented in Appendix A.

### 4.7. Measurement of Circulating Levels of Adipokines

Human serum fatty acid-binding protein (FABP)-4 (Intra-assay Variability: 2.5%, Inter-assay Variability: 3.9%, sensitivity: 0.08 ng/mL) and fibroblast growth factor (FGF)-21 (Intra-assay Variability: 2%, Inter-assay Variability: 3.3%, sensitivity: 7 pg/mL), were measured using commercially available immunoassays from Biovendor, Brno, Czech Republic and human Adiponectin (Intra-assay Variability: 2.3%, Inter-assay Variability: 5.4%, sensitivity: 1.25 ng/mL) from Mercodia, Uppsala, Sweden. Rat FABP-4 ELISA Kit was purchased from Boster Biological Technology Co., Ltd. (Pleasanton, CA, USA) (Intra-assay Variability: 5.2%, Inter-assay Variability: 5.3%, sensitivity: <15 pg/mL). Measurements were performed in duplicate and any sample with a coefficient of variation >20% was repeated or excluded.

### 4.8. Statistical Analysis

Statistical analyses were performed with GraphPad Prism version 5.03 (GraphPad Software, La Jolla, CA, USA). One-way ANOVA with Tukey’s posttest was used when comparing multiple groups or Student’s *t*-test when comparing two groups. Correlation of circulating FGF-21 and FABP-4 levels, as well as adiponectin levels was analyzed by non-parametric Spearman test. Data are presented as mean and standard error of the mean (SEM). Differences were considered statistically significant if *p* < 0.05.

## 5. Conclusions

This is the first study in which the adipokine and metabolic gene expression profile of adipose tissue has been examined in the three most commonly used rodent models of pulmonary hypertension. The novel concept put forward here is that altered adipose tissue bioenergetics are a component of experimental pulmonary hypertension that may be amenable to therapeutic targeting.

## Figures and Tables

**Figure 1 ijms-22-01435-f001:**
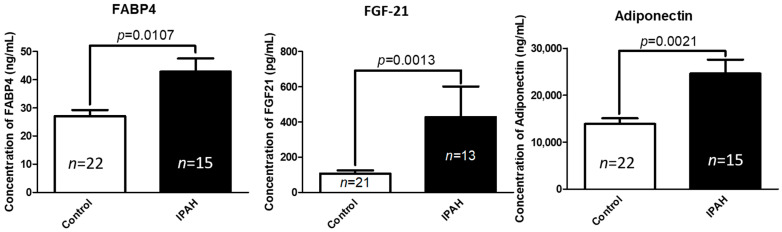
Circulating plasma fatty acid binding protein 4 (FABP-4), fibroblast growth factor -21 (FGF-21), and adiponectin levels are elevated in patients with IPAH compared to controls. Data represent mean ± SEM from *n* = 15–22 subjects per group (see Appendix A for patient characteristics). Control: Normal controls, IPAH: Idiopathic Pulmonary Arterial Hypertension. Statistical analysis by Student’s *t*-test.

**Figure 2 ijms-22-01435-f002:**
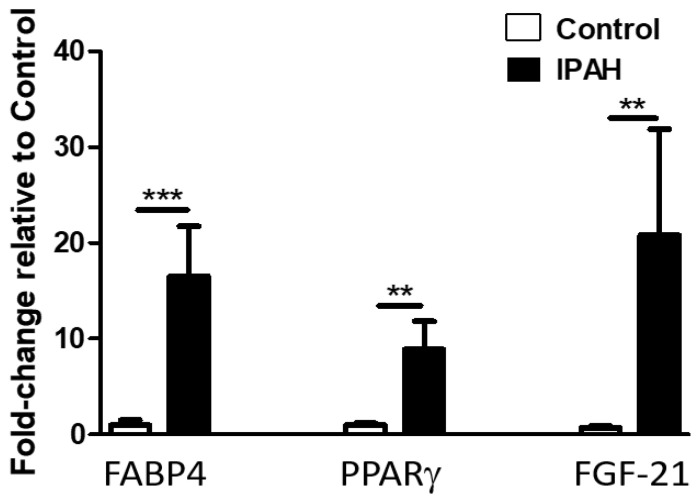
Increased Lung levels of FABP-4, peroxisome proliferator-activated receptor γ (PPARγ), and FGF-21 mRNA in human ΙPAH compared to controls. Data represent mean ± SEM from *n* = 15–22 patients per group (see Appendix A for patient characteristics). Statistical analysis by Student’s *t*-test. ** *p* < 0.005, *** *p* < 0.001.

**Figure 3 ijms-22-01435-f003:**
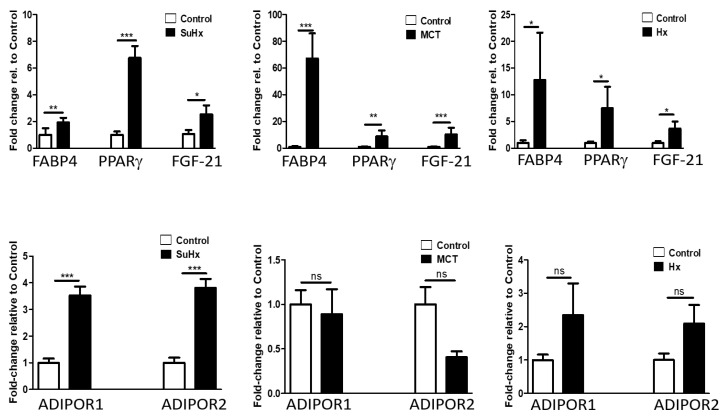
Altered Lung levels of FABP-4, PPARγ, FGF-21, and adiponectin receptor 1 and 2 mRNA in experimental pulmonary hypertension. Data represent mean ± SEM from *n* = 5–12 animals per experimental group. Statistical analysis by Student’s *t*-test. * *p* < 0.05, ** *p* < 0.005, *** *p* < 0.001, ns: no significant.

**Figure 4 ijms-22-01435-f004:**
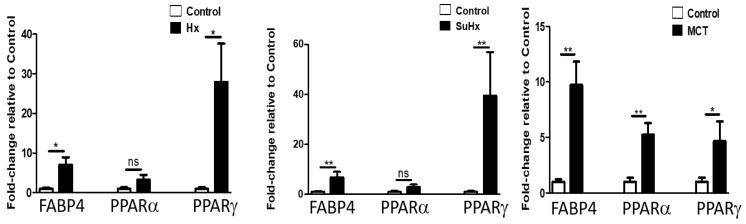
Altered adipose tissue levels of FABP-4, PPARα and PPARγ mRNA in experimental pulmonary hypertension. Data represent mean ± SEM from *n* = 8–14 animals per experimental group. Statistical analysis by Student’s *t*-test. * *p* < 0.05, ** *p* < 0.005, ns: non significant.

**Figure 5 ijms-22-01435-f005:**
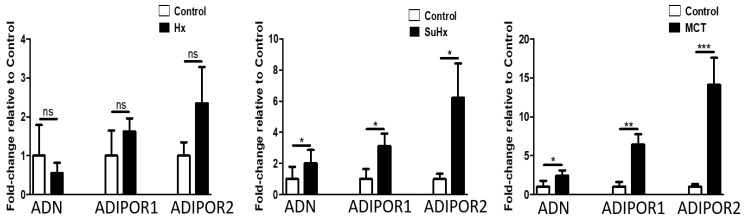
Altered adipose tissue levels of Adiponectin and Adiponectin receptor 1 and 2 mRNA in experimental pulmonary hypertension. Data represent mean ± SEM from *n* = 8–14 animals per experimental group. Statistical analysis by Student’s *t*-test. * *p* < 0.05, ** *p* < 0.005, *** *p* < 0.001, ns: no significant.

**Figure 6 ijms-22-01435-f006:**
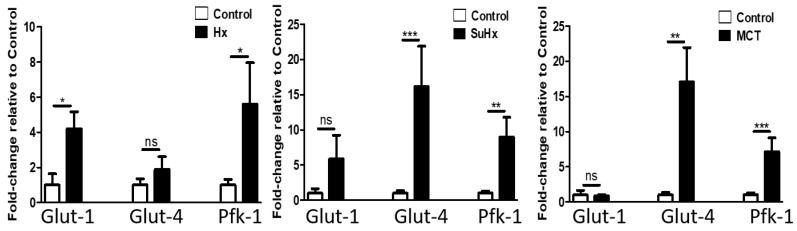
Altered adipose tissue levels of glycolytic marker mRNA in experimental pulmonary hypertension. Data represent mean ± SEM from *n* = 8–14 animals per experimental group. Statistical analysis by Student’s *t*-test. * *p* < 0.05, ** *p* < 0.005, *** *p* < 0.001, ns: no significant.

**Figure 7 ijms-22-01435-f007:**
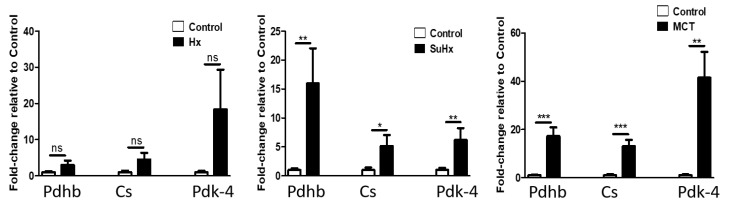
Altered adipose tissue levels of tricarboxylic acid cycle (TCA) marker mRNA in experimental pulmonary hypertension. Data represent mean ± SEM from *n* = 8–14 animals per experimental group. Statistical analysis by Student’s *t*-test. * *p* < 0.05, ** *p* < 0.005, *** *p* < 0.001, ns: no significant.

**Figure 8 ijms-22-01435-f008:**
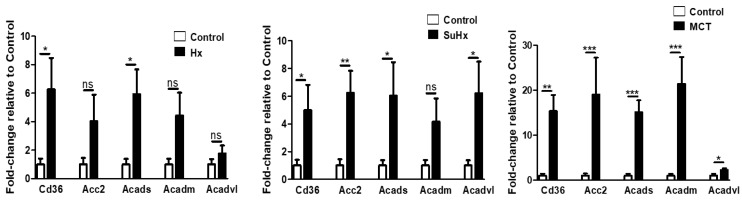
Altered adipose tissue levels of fatty acid oxidation marker mRNA in experimental pulmonary hypertension. Data represent mean ± SEM from *n* = 8–14 animals per experimental group. Statistical analysis by Student’s *t*-test. * *p* < 0.05, ** *p* < 0.005, *** *p* < 0.001, ns: no significant.

**Table 1 ijms-22-01435-t001:** Hemodynamic characteristics of experimental pulmonary hypertension. Right Ventricular Systolic Pressure (RVSP), Left Ventricular Systolic Pressure (LVSP), Fulton’s Index (FI).

	RVSP (Mean ± SEM) mm Hg	LVSP (Mean ± SEM) Mm Hg	Fulton’s Index (Mean ± SEM)	RV/BW Ratio (Mean ± SEM)
Normoxia	21.20 ± 1.8	81.83 ± 6.4	0.24 ± 0.006	0.55 ± 0.02
Hypoxia (Hx)	37.03 ± 1.5 *	84.43 ± 6.6	0.40 ± 0.050 *	0.88 ± 0.09 *
Monocrotaline (MCT)	52.18 ± 8.8 *	97.11 ± 1.9	0.46 ± 0.035 *	0.94 ± 0.09 *
SuHx	38.91 ± 2.3 *	69.83 ± 6.6	0.69 ± 0.048 *	1.69 ± 0.14 *

Data represent mean ± SEM from *n* = 5–16 animals per experimental group. Statistical analysis by Student’s *t*-test. * *p* < 0.05 compared to normoxia control.

## Data Availability

The data presented in this study are available in the article or Appendix A.

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
