# Peer review of "Adipokines and Metabolic Regulators in Human and Experimental Pulmonary Arterial Hypertension"

_ijms, 2021, doi:10.3390/ijms22031435_

Round 1

Reviewer 1 Report

General Comments

This is a well written and logically presented study that combines outcomes from human clinical specimens and rat experimental models to evaluate the expression of adipose derived adipokines and biomarkers in the context of pulmonary hypertensive disease.  The data are presented in a clear and quantitative manner with appropriate attention to statistical outcomes.  The results are discussed with emphasis on both their supported conclusions and their potential limitations.  In general, the authors have done an excellent job in citing the relevant recent literature; however, there are potentially additional papers that might merit consideration for inclusion in the Discussion (see below):

Int J Mol Sci 2017 Dec 28;19(1):73 Zhou Y et al. Omentin-A Novel Adipokine in Respiratory Diseases

Pulm Circ 2020 Sep 18;10(3):2045894020952019.  Mair KM et al.  Obesity, estrogens and adipose tissue dysfunction - implications for pulmonary arterial hypertension

Resistin family proteins in pulmonary diseases.

Lin Q, Johns RA.Am J Physiol Lung Cell Mol Physiol. 2020 Sep 1;319(3):L422-L434.

Specific Comments

Ln 51.  In this paragraph introducing the study, the authors should specify the specie(s) of interest to their experimental model.  As written, a reader is led to believe that the paper is about to describe work in human subjects and/or models.  The specie of the experimental pulmonary hypertension model should be specified.

Ln 83.  Define Su/Hx when first introduced into the text.

Ln 84.  Define MCT when first introduced.

Ln 323.  Change “3” to “three”.

Ln 342.  Change PPARg to PPARγ.

Ln 351.  Additional confounders to be considered are the body mass index (BMI) and diabetes status of the subjects.

Author Response

General Comments 

This is a well written and logically presented study that combines outcomes from human clinical specimens and rat experimental models to evaluate the expression of adipose derived adipokines and biomarkers in the context of pulmonary hypertensive disease. The data are presented in a clear and quantitative manner with appropriate attention to statistical outcomes. The results are discussed with emphasis on both their supported conclusions and their potential limitations. In general, the authors have done an excellent job in citing the relevant recent literature; however, there are potentially additional papers that might merit consideration for inclusion in the Discussion (see below):

We thank the reviewer for recognizing the importance of our work and for recommending additional citations which we have now included in the revised version of our manuscript (Citations 5,12 and 25)

Specific Comments

Ln 51. In this paragraph introducing the study, the authors should specify the specie(s) of interest to their experimental model. As written, a reader is led to believe that the paper is about to describe work in human subjects and/or models. The specie of the experimental pulmonary hypertension model should be specified.

We now specify that our experimental models are in rats (line 55 of the revised manuscript)

Ln 83. Define Su/Hx when first introduced into the text.

We made this edit per the reviewer’s suggestion (line 84 in revised manuscript)

Ln 84. Define MCT when first introduced.

We made this edit per the reviewer’s suggestion (line 85 in revised manuscript)

Ln 323. Change “3” to “three”.

We  made this change (line 327 in revised manuscript)

Ln 342. Change PPARg to PPARγ.

We have changed PPARg to PPARγ throughout the manuscript and figures

Ln 351. Additional confounders to be considered are the body mass index (BMI) and diabetes status of the subjects.

We have added these confounders in the revised discussion per the reviewer’s suggestion (line 358)

Reviewer 2 Report

The manuscript is focused on evaluating the circulating level of adiponectin, FGF-21 and FABP4 in idiopathic pulmonary arterial hypertension (IPAH), and mRNA level of these molecules together with PPARgamma in lungs of patiens with IPAH. Then the authors measure the same parameters in three commonly used rat models of pulmonary hypertension, i.e. hypoxia, sugen-hypoxia and monocrotaline models, and observe the same trend. The authors conclude that metabolic changes observed in human PAH are recapitulated in experimental PH, and maybe causative to this disease. 

The data obtained in human IPAH samples is indeed very interesting. However, it is not clear to me the relevance of hypoxia model and partially sugen-hypoxia model to IPAH? Why use these two models, as they are more relevant to PH Group 3 than to PH group 1, which is IPAH. Indeed, the degree of observed changes is different across all three models (Figure 3), and hypoxia model stands out from the other two (Figures 5 and 6). Besides, hypoxia alone could have been at least done for three weeks, as sugen-hypoxia, to allow for more comparable analysis. On the other hand, while rat monocrotaline model is commonly used as a Group 1 PAH model, it is pro-inflammatory so that it would be not very objective towards an increase of some of the parameters measured. So how do you distinguish the changes between the three models and differentiate between the human and rat data? These shortfalls should be mentioned and discussed in more details. 

Regarding the figures' presentation, it would be certainly better to see all the graphs presented as dot-plots. In some figures, the numbers vary between 8 and 14, or even worse, between 5 and 12 rats per group. This is not ideal, so please explain why such big differences were used and how exactly you did your power calculation to come to those numbers? 

Use of reverse transcription to study gene expression is a bit outdated approach, which could have been justified if the authors employed it to verify gene expression data obtained in an unbiased gene array or metabolic (metabolomics) screening. 

Interestingly, the authors observed the differences in PAPRgamma, FABP4 and FGF21 mRNA expression, but what about the functionally-relevant protein? Was it measured? In other words, did mRNA changes translated into the functional changes? In the discussion, the author mentioned that these changes were not present in pulmonary vessels, and suggest that other (extra-pulmonary) cells contribute to this phenomenon. Do the authors know what cells that could be? Inflammatory cells or adipocytes? How about perivascular fat, which recently has been considered as an important regulator of systemic vascular function? 

Another question is related to BMI in IPAH group – is it possible that the changes that the authors see are simply reflecting the increased body mass and BMI in a disease group. Can perivascular fat be determined? Can the authors measure adipocyte size or/and number, maybe from the histological human lung samples? Also, having animal models provide with the opportunity to measure adipose tissue level, for example, if it is different between the groups, and look at adipocytes more closely.

Finally, the novelty of the study is not very clear and hence is questionable, especially considering that there are many publications and reviews in this area. Please state very clearly what the novelty is. 

Author Response

Comments and Suggestions for Authors

1. The manuscript is focused on evaluating the circulating level of

adiponectin, FGF-21 and FABP4 in idiopathic pulmonary arterial

hypertension (IPAH), and mRNA level of these molecules together

with PPARgamma in lungs of patiens with IPAH. Then the authors

measure the same parameters in three commonly used rat models

of pulmonary hypertension, i.e. hypoxia, sugen-hypoxia and

monocrotaline models, and observe the same trend. The authors

conclude that metabolic changes observed in human PAH are

recapitulated in experimental PH, and maybe causative to this

disease.

The data obtained in human IPAH samples is indeed very

interesting. However, it is not clear to me the relevance of hypoxia

model and partially sugen-hypoxia model to IPAH? Why use these

two models, as they are more relevant to PH Group 3 than to PH

group 1, which is IPAH. Indeed, the degree of observed changes is

different across all three models (Figure 3), and hypoxia model

stands out from the other two (Figures 5 and 6). Besides, hypoxia

alone could have been at least done for three weeks, as sugenhypoxia,

to allow for more comparable analysis. On the other

hand, while rat monocrotaline model is commonly used as a Group

1 PAH model, it is pro-inflammatory so that it would be not very

objective towards an increase of some of the parameters

measured. So how do you distinguish the changes between the

three models and differentiate between the human and rat data?

These shortfalls should be mentioned and discussed in more

details.

We thank the reviewer for raising these very important points about the different pathogenetic mechanisms among rodent models and about the relevance of these models to the different Groups of human PAH. We agree with the reviewer that all models have limitations and we have added a segment in the discussion that addresses these shortfalls (discussion-lines 362-366). We have also included additional citations which describe the limitations of existing models (61,62). We believe our results in IPAH patients are indeed interesting and perhaps our data in preclinical models can be useful in further exploring common pathophysiologic mechanisms between the different groups of human PAH and also in developing novel therapeutics for all groups.

2. Regarding the figures' presentation, it would be certainly better to

see all the graphs presented as dot-plots. In some figures, the

numbers vary between 8 and 14, or even worse, between 5 and

12 rats per group. This is not ideal, so please explain why such big

differences were used and how exactly you did your power

calculation to come to those numbers?

We acknowledge that there is variation in the number of animals per experimental group and this is simply due to technical factors (including availability and quality of biological samples from each control and experimental group). We did not perform power calculations for these experimental groups because we evaluated multiple outcomes and we did not have any information to guide assumptions about the expected differences for each variable examined among the experimental groups. Similarly, we presented our data in standard bar graphs by depicting the mean and SEM because due to the large number of outcomes, we felt that differences would be harder to discern in dot-plots.

3. Use of reverse transcription to study gene expression is a bit

outdated approach, which could have been justified if the authors

employed it to verify gene expression data obtained in an

unbiased gene array or metabolic (metabolomics) screening.

We opted for a targeted approach with determination of abundance of the specific mRNA transcripts by Real time PCR but we agree with the reviewer that more comprehensive insights into gene expression and metabolic regulation can be gained from high throughput studies such as RNAseq and metabolomics. These studies are beyond the scope of the current manuscript.

4. Interestingly, the authors observed the differences in

PAPRgamma, FABP4 and FGF21 mRNA expression, but what

about the functionally-relevant protein? Was it measured? In other

words, did mRNA changes translated into the functional changes?

We thank the reviewer for raising this valid point. We did not perform Western analysis in these studies due to a limited amount of biological samples and we acknowledge this as a limitation in the revised manuscript (Discussion, line 355).

5. In the discussion, the author mentioned that these changes were

not present in pulmonary vessels, and suggest that other (extrapulmonary)

cells contribute to this phenomenon. Do the authors

know what cells that could be? Inflammatory cells or adipocytes?

This is a very intriguing question and the reviewer is correct, the cellular sources of these metabolic regulators are a very important topic for further study. We would like to clarify that our comment only pertains to FABP4 in human lungs because we evaluated its expression in pulmonary arteries and found no differences between IPAH and controls so we suggested that extravascular sources (not extrapulmonary) may be the source. The reviewer is correct, inflammatory cells, adipocytes or other cells in the lung may be the source of FABP4 and the best way to address this question would be to perform single cell RNAseq in future studies.

6. How about perivascular fat, which recently has been considered

as an important regulator of systemic vascular function?

This is a very important point, unfortunately we are not able to measure perivascular fat in these studies since we do not have access to fixed tissue.

7. Another question is related to BMI in IPAH group – is it possible

that the changes that the authors see are simply reflecting the

increased body mass and BMI in a disease group. Can

perivascular fat be determined? Can the authors measure

adipocyte size or/and number, maybe from the histological human

lung samples? Also, having animal models provide with the

opportunity to measure adipose tissue level, for example, if it is

different between the groups, and look at adipocytes more closely.

As shown in Table A1 there were no differences in BMI between the control and IPAH group we therefore so not believe that the observed differences in adipokine levels reflect increased BMI in the disease group. We thank the reviewer for the great suggestions but unfortunately we do not have access to fixed tissue from these patients so we cannot measure perivascular fat or adipocyte size/number. We agree that the preclinical models better lend themselves to these studies and we will plan to address these important questions in future studies. 

Reviewer 3 Report

Major comment

The present study demonstrated the changes in the expression of genes that are related to metabolism in human and experimental pulmonary hypertension. Except for the plasma level of FABP3, FGF-21 and adiponectin in the patients with pulmonary hypertension, the mRNA expression was investigated. The present study is purely descriptive study, while pathophysiological relevance of the findings and/or the underlying mechanisms remains unaddressed. Therefore, the present study is judged to be preliminary.

Specific points and minor points

1. The Discussion section contains many speculations. 

2. Reference 4 is not appropriate to support the statement on lines 34-37. Reference 4 demonstrated the higher incidence of insulin resistant in the pulmonary hypertension female patients compared to the control. Furthermore, the study reported that there was no difference in body mas index between pulmonary hypertension and control, while it did no investigate the association between IPAH and metabolic disorder.

2. "gamma (in Greek)" is needed after PPAR (line 95)

3. Figure 2: NC of the label of the ordinate should be defined.

Author Response

Major comment

The present study demonstrated the changes in the expression of genes that are related to metabolism in human and experimental pulmonary hypertension. Except for the plasma level of FABP3, FGF-21 and adiponectin in the patients with pulmonary hypertension, the mRNA expression was investigated. The present study is purely descriptive study, while pathophysiological relevance of the findings and/or the underlying mechanisms remains unaddressed. Therefore, the present study is judged to be preliminary.

We have acknowledged these limitations of our study in the discussion.

Specific points and minor points

  1. The Discussion section contains many speculations

We acknowledge that our study provides a set of observations that need to be further explored with mechanistic studies in the different experimental models of pulmonary hypertension to define the contribution of adipose tissue-derived cytokines and bioenergetics to disease pathogenesis, progression and response to treatments. We have included several statements in the discussion in an effort to interpret our findings in the context of the existing literature and we have revised the discussion to better indicate that many of these statements are indeed speculative (lines 225, 235, 287)

  1. Reference 4 is not appropriate to support the statement on lines 34-37. Reference 4 demonstrated the higher incidence of insulin resistant in the pulmonary hypertension female patients compared to the control. Furthermore, the study reported that there was no difference in body mas index between pulmonary hypertension and control, while it did no investigate the association between IPAH and metabolic disorder.

We thank the reviewer for pointing out the need to add references that more clearly describe the association between obesity, metabolic syndrome and PAH.  In addition to reference 4 that described insulin resistance in pulmonary hypertension female patients, we now cite a review article by Mair et al (citation 5), and a study by Poms et al (REVEAL registry analysis, citation 6) in which the presence of obesity, diabetes and hypertension were associated with worse outcomes in pulmonary arterial hypertension.  Two review articles, by Assad et al (citation 7) about metabolic dysfunction in PAH and by Friedman et al (citation 8) about obesity and pulmonary hypertension were also added.

  1. "gamma (in Greek)" is needed after PPAR (line 95)

We have corrected this to PPARg

  1. Figure 2: NC of the label of the ordinate should be

We have corrected the ordinate label to control in Figure 2 and all other figures

Reviewer 4 Report

Pulmonary arterial hypertension is a progressive disease with high mortality. Although, current approved therapies showed improvements in quality of life and hemodynamic parameters, they have demonstrated only very limited beneficial effects on survival and disease progression. Thus, there is a pressing unmet need for more effective treatments of pulmonary arterial hypertension. A better understanding of the PAH pathogenesis will provide the basis for novel therapeutic approaches that may confer benefit in these patients.

In this manuscript, the authors investigated a panel of primarily adipose tissue-derived or therein implicated metabolic regulators in human and experimental pulmonary hypertension and evaluated the metabolic state of adipose tissue by assessing mRNA levels of genes involved in glycolysis, the tricarboxylic acid cycle and fatty acid oxidation in experimental pulmonary hypertension. Overall, the manuscript is well written. The study is well designed and results are straightforward. The methods are well described and comply with reporting standards. The conclusions appear to be supported by the study results. However, there are some issues to be addressed prior to acceptance of the manuscript for publication.

Major

If the control group is the same for all groups, why are the expression data for the experimental groups presented separately?

If the control group is the same for all groups, why was the Student t test applied and not the ANOVA with Dunnett's test?

Why were the circulating levels of FGF-21 and adiponectin not measured in rats?

Line 199. Expression of FGF-21 mRNA in adipose tissue are not presented in the results.

Lines 283-285. The species-specific differences might not account for the discrepancy because, in the current study, the FGF-21 mRNA expression in rat lungs was enhanced.

Minor

Figures 2-4. Please correct in the labeling “PPAR-y” to “PPARγ”

Figures 2-4. Please consistency in the labeling of the control group and use either “NC” or “Control”

Figures 4. Results of PPARα mRNA expression are not described in the text.

Table A2. Please use decimal points instead of commas as decimal separators (values for leukocyte numbers).

Line 88. Please define the abbreviation RV/BW

Line 95. Please correct “PPAR” to “PPARγ”

Abbreviations should be defined at first mention in the text only once (e.g., for FAO lines 57, 114 and 130; TCA line 57, 113 and 123; RVSP lines 80 and 403; LVSP lines 85 and 403; FI lines 88, 414, 416; PDK lines 126 and 129; ACAD lines 133 and 320).

No need to introduce abbreviations if the term is used less than 3 times in the text (PDHb lines 123 and 126; PDK lines 126 and 129; CS line 133; S, M and VL lines 133 and 134; ACAD lines 133 and 320; ACC-2 line 137; PAECs line 282; PAs line 300; OXPHOS line 306; BMPR2 line 314; ).

Line 342. Please correct in the labeling “PPARg” to “PPARγ”

Author Response

Pulmonary arterial hypertension is a progressive disease with high mortality. Although, current approved therapies showed improvements in quality of life and hemodynamic parameters, they have demonstrated only very limited beneficial effects on survival and disease progression. Thus, there is a pressing unmet need for more effective treatments of pulmonary arterial hypertension. A better understanding of the PAH pathogenesis will provide the basis for novel therapeutic approaches that may confer benefit in these patients. In this manuscript, the authors investigated a panel of primarily adipose tissue-derived or therein implicated metabolic regulators in human and experimental pulmonary hypertension and evaluated the metabolic state of adipose tissue by assessing mRNA levels of genes involved in glycolysis, the tricarboxylic acid cycle and fatty acid oxidation in experimental pulmonary hypertension. Overall, the manuscript is well written. The study is well designed and results are straightforward. The methods are well described and comply with reporting standards. The conclusions appear to be supported by the study results. However, there are some issues to be addressed prior to acceptance of the manuscript for publication.

We thank the reviewer for recognizing the importance of our work. 

Major comments: 

1. If the control group is the same for all groups, why are the expression data for the experimental groups presented separately?

The control groups were not the same for all experimental groups. A separate control group was used for each experimental model using a group of animals that were age-matched and studied concurrently with the three experimental groups. As described in the methods section, the control group for the hypoxic model was a normoxic group of the same age, housed in the animal facility and sacrificed at the same time as the experimental group (2 weeks). The control group for the MCT model received an injection of vehicle (PBS) and the animals were sacrificed on day 28. The control group for the SuHx model was a normoxic group that received a vehicle injection (DMSO) and the animals were sacrificed on day 24. For simplicity of labelling we designated each control group as ‘control’ in the  figures and we added  labelling of the  y axes in the revised figures for further clarity. 

2. If the control group is the same for all groups, why was the Student t test applied and not the ANOVA with Dunnett's test?

The control group was not the same for all groups. These separate control groups were age-matched and studied concurrently with the experimental animals so it would not be appropriate to compare them using ANOVA. 

3. Why were the circulating levels of FGF-21 and adiponectin not measured in rats?

We had very limited amount of rat plasma and thus were not able to measure FGF-21 or adiponectin in our rat experiments

4. Line 199. Expression of FGF-21 mRNA in adipose tissue are not presented in the results.

We thank the reviewer for pointing this oversight on our part and we apologize. Indeed we measured FGF-21 mRNA in adipose tissue and did not find statistically significant changes in FGF-21 mRNA in any of the experimental models studied. We have edited the results and discussion sections accordingly (lines112 and 204).

5. Lines 283-285. The species-specific differences might not account for the discrepancy because, in the current study, the FGF-21 mRNA expression in rat lungs was enhanced.

We agree with the reviewer and modified our statement.

Minor comments: 

1. Figures 2-4. Please correct in the labeling “PPAR-y” to “PPARγ”

We have corrected this

2. Figures 2-4. Please consistency in the labeling of the control group and use either “NC” or “Control”

We thank the reviewer for pointing out the discrepancy in our labelling. We have changed the labeling to “Control” in all figures

3. Figures 4. Results of PPARα mRNA expression are not described in the text.

We have added a description of these results in the revised manuscript  (line 107)

4. Table A2. Please use decimal points instead of commas as decimal separators (values for leukocyte numbers).

We have corrected this per the reviewer’s suggestion.

5. Line 88. Please define the abbreviation RV/BW

We have defined RV/BW as Right ventricular weight/Body weight ratio per the reviewer’s suggestion

6.Line 95. Please correct “PPAR” to “PPARγ”

We have corrected this

7.Abbreviations should be defined at first mention in the text only once (e.g., for FAO lines 57, 114 and 130; TCA line 57, 113 and 123; RVSP lines 80 and 403; LVSP lines 85 and 403; FI lines 88, 414, 416; PDK lines 126 and 129; ACAD lines 133 and 320).

We made these edits in the revised manuscript per the reviewer’s suggestion

8. No need to introduce abbreviations if the term is used less than 3 times in the text (PDHb lines 123 and 126; PDK lines 126 and 129; CS line 133; S, M and VL lines 133 and 134; ACAD lines 133 and 320; ACC-2 line 137; PAECs line 282; PAs line 300; OXPHOS line 306; BMPR2 line 314; ).

 We have made these edits in the revised manuscript per the reviewer’s suggestion

9.Line 342. Please correct in the labeling “PPARg” to “PPARγ”

 We have corrected this

Round 2

Reviewer 2 Report

The authors addressed the questions, but have not provided any additional data or figure presentation. I agree that they cannot improve the manuscript further at this point.

Author Response

All questions were previously addressed 

Reviewer 3 Report

The authors made some minor improvement. However, the major drawback, i.e., the lack of mechanistic insight into the observations and the lack of establishment of pathophysiological relevance of the observations, still remains. The revised study is still judged to be preliminary.

Author Response

All questions were previously addressed

Reviewer 4 Report

I have no further comments. The authors addressed al the concerns. I recommend acceptance of the manuscript in its present form.

Author Response

All questions were previously addressed